# Knowledge Graph Retrieval-Augmented Generation via GNN-Guided Prompting

**Haochen Liu, Song Wang, Jundong Li**
University of Virginia
{sat2pv,sw3wv,jundong}@virginia.edu

## Abstract

Large Language Models (LLMs) have demonstrated remarkable performance in open-domain question answering (QA), but their reliance on knowledge learned during pretraining limits their ability to provide accurate and up-to-date information. Knowledge Graph Retrieval-Augmented Generation (KG-RAG) enhances LLMs by incorporating structured knowledge from knowledge graphs (KGs). A common approach in KG-RAG is to retrieve relevant knowledge paths starting from entities in the input question and expanding along KG edges by LLM reasoning. However, existing KG-RAG methods suffer from the challenge that retrieval is performed step by step greedily using only local graph context, which can lead to retrieval errors that prematurely discard essential paths. To address the issue and perform more accurate retrieval, we propose **GGR** (**G**NN-**G**uided **R**etrieval for LLM Reasoning), a novel GNN-enhanced KG-RAG framework that integrates graph-based relevance scoring into the retrieval process. Our approach computes global importance scores across a contextualized subgraph, ensuring that key reasoning knowledge paths are preserved, even if their local relevance appears weak. Additionally, we introduce local semantic alignment by incorporating query-relation semantic similarity, refining the relation selection of LLM. Extensive experiments on Question-Answering tasks demonstrate that our method significantly improves retrieval accuracy and answer quality, demonstrating the effectiveness of combining graph-based reasoning and LLM-driven retrieval for structured knowledge integration.[1]

## 1 Introduction

Large Language Models (LLMs) (Brown et al., 2020; Touvron et al., 2023) have achieved remarkable success in natural language processing (NLP) tasks, including text comprehension (Lewis et al., 2020a), language generation (Cheng et al., 2023), and open-domain question answering (QA) (Wei et al., 2022; Cohen et al., 2024; Chen et al., 2024). However, their reliance on knowledge learned during pretraining limits their ability to handle tasks requiring factual accuracy, dynamic knowledge updates, and multi-hop reasoning (Zheng et al., 2023; Wang et al., 2023). In these tasks, LLMs struggle with outdated information, incomplete knowledge, and unverifiable claims. A typical approach to overcome these challenges is to enable LLMs to access and reason over external sources of knowledge to improve their reliability (Jiang et al., 2024; Sun et al., 2024).

To address the limitations of LLMs in knowledge-intensive tasks where external knowledge is needed, Retrieval-Augmented Generation (RAG) (Gao et al., 2023) enhances LLMs by retrieving external knowledge sources such as Knowledge Graphs (KGs). Knowledge Graph RAG (KG-RAG) (Sanmartin, 2024) has gained attention for its structured and rich knowledge representation, improving interpretability and factual consistency of the model's responses. Most KG-RAG methods for LLM generation follow a two-step process: (1)

---

[1]Our code is here: https://github.com/HaochenLiu2000/GGR.

**Knowledge Retrieval**, where relevant entities and relations in the KG are extracted based on the input query, and (2) **Prompt Integration**, where retrieved knowledge from the KG is incorporated into the LLM's input to refine its response. Typically, KG-RAG systems start from query entities and expand along relational paths, retrieving multiple query-relevant triplets as prompt evidence for LLM reasoning. To select helpful knowledge for the query, LLMs are often used to evaluate and select triplets via similarity scoring or prompt-based reasoning (Sun et al., 2024; Ma et al., 2024).

Despite the integration of knowledge graphs, during the retrieval process, existing KG-RAG methods often fail to capture and utilize the global information in the knowledge graph, i.e., KG information beyond a node's immediate neighborhood that possibly participates in long-range reasoning chains leading to the answer, as shown in Figure 1. In particular, current approaches typically perform stepwise greedy retrieval, relying solely on local knowledge at each step (Sun et al., 2024). Without a global view of the graph structure, these methods are prone to retrieval errors, where seemingly weak but structurally important paths are prematurely discarded, disrupting multi-hop reasoning.

To address the limitation of lacking global information during retrieval, we propose GGR (**G**NN-**G**uided **R**etrieval for LLM Reasoning), a novel KG-RAG framework that incorporates a graph neural network (GNN) into the retrieval process to reason on global graph information. The GNN serves as an

Figure 1: The figure illustrates the limitation of lacking global views in existing KG-RAG LLM retrieval. Green entity is in the question, and purple entity is the correct answer. After retrieving *milk*, the LLM compares the question with *promote* and *contain*. The LLM tends to select *promote*, which is seemingly more relevant to the question, leading to a wrong path for its lack of global KG information, causing the correct reasoning path to be pruned early. Notably, typical methods first select relations and then identify corresponding entities to reduce computational overhead.

auxiliary scorer to assess question-aware relevance over the graph for the LLMs. However, there exist two difficulties to overcome. First, to present the global information of the relevance of knowledge in a compact form for LLM reasoning, we introduce a GNN scoring mechanism that evaluates the relevance of entities and relations within a contextualized subgraph. By propagating information through the graph, the GNN captures structural dependencies beyond local neighborhoods, allowing globally important reasoning paths to be preserved. Second, to avoid excessive influence from purely global relevance signals, we further incorporate query-relation semantic similarity that assesses how well a relation semantically matches the intent of the question. Although GNN scores provide global guidance, relation similarity offers local semantic precision. By integrating these two signals, GGR enables more reliable and context-aware knowledge retrieval, effectively overcoming the limitation of existing KG-RAG methods. Our main contributions are as follows:

- We analyze the limitations of existing KG-RAG methods, highlighting the issues of stepwise greedy retrieval errors and difficulties for incorporating global KG information in the retrieval process.

- We propose a novel retrieval framework that combines GNN-based global relevance scoring with query-relation semantic similarity. This mechanism enables structurally aware and semantically precise knowledge selection, mitigating incorrect knowledge path pruning and reasoning errors.

- Through extensive experiments on knowledge-intensive QA tasks, we show that our method significantly surpasses current state-of-the-art methods, demonstrating the benefits of combining graph reasoning and LLM-driven retrieval in the tasks of KG-RAG. Experiments also show that even simple GNN architectures suffice for strong performance in KG-RAG.

## 2    Problem Formulation

In this section, we formally define the task of Knowledge Graph Retrieval-Augmented Generation (KG-RAG) for LLMs. In KG-RAG, structured knowledge from a knowledge graph is retrieved and integrated into the generation process. A knowledge graph is represented as $\mathcal{G} = (\mathcal{E}, \mathcal{R}, \mathcal{T})$, where $\mathcal{E}$ is the set of entities, $\mathcal{R}$ is the set of relations, and $\mathcal{T} = \{(h, r, t) \mid h, t \in \mathcal{E}, r \in \mathcal{R}\}$ is the set of knowledge triplets. Each triplet consists of a head entity $h$, a relation $r$, and a tail entity $t$. Given an LLM, denoted as $LM$, and a question $q$, the goal of KG-RAG is to retrieve relevant knowledge from $\mathcal{G}$ and use it as additional information for the LLM to improve its response generation for $q$.

## 3    GNN-Guided Retrieval for LLM Reasoning

In this section, we introduce GGR, a GNN-enhanced KG-RAG framework that improves knowledge retrieval for LLMs by integrating KG information and graph reasoning into the retrieval process. As shown in Figure 2, our approach consists of three key phases: *(i)* **Subgraph Extraction**, where we construct a task-relevant contextualized subgraph from the knowledge graph based on entities in the question; *(ii)* **Question-Relevance Scoring**, where we use GNNs and the query-relation semantic similarity to assign scores to entities and relations to show their relevance to the query. This hybrid scoring mechanism captures both global graph structure and local semantic alignment to guide the retrieval. and *(iii)* **LLM Stepwise Retrieval**, where knowledge is retrieved in a stepwise manner, selecting triplets by LLMs with the assistance of both GNN scores and query-relation semantic similarity scores to address reasoning bias. The retrieved knowledge is then formatted into prompts of LLMs to generate a response. In the following subsections, we describe each step in detail.

### 3.1    Subgraph Extraction

To effectively retrieve relevant knowledge from the KG for a given question, we extract a contextualized subgraph that captures helpful structured information while reducing the search space. Specifically, for a given question $q$, we first identify the set of entities in $\mathcal{G}$ that explicitly appear in $q$, denoted as $\mathcal{E}_q$. Using a predefined hop limit $N$, we then extract the $N$-hop neighbors of these entities along with their connecting edges to form the contextualized subgraph for $q$, denoted as $\mathcal{G}_q$ (Yasunaga et al., 2022a). We define $\mathcal{T}_q$ as the set of triplets in subgraph $\mathcal{G}_q$. This subgraph encapsulates potentially useful knowledge for the LLM in generation and serves as the input for the subsequent GNN scoring phase.

### 3.2    Question-Relevance Scoring

With the contextualized subgraph $\mathcal{G}_q$, the next step is to assess the relevance of each entity and relation to the question $q$ before retrieval as auxiliary information for the LLMs. Traditional KG-RAG methods rely solely on LLM reasoning to evaluate local relevance during stepwise expansion, which can lead to pruning of important multi-hop reasoning paths due to the lack of global KG information. To address this, we adopt a hybrid scoring strategy that combines two complementary signals: (1) GNN Scores: Global structural relevance scores provided by a graph neural network (GNN), which propagates knowledge information across the subgraph, and (2) Semantic Similarity Scores: Local semantic similarity measured by computing the embedding similarity between the question and relations. These scores guide the subsequent retrieval process, allowing the model to retain structurally important and semantically aligned knowledge.

**GNN Scores.**    Each entity $e \in \mathcal{E}_q$ and relation $r \in \mathcal{R}_q$ in the subgraph are initialized with embeddings. The entity embeddings can either be obtained from a pretrained KG embedding model or generated dynamically using an LLM. The relation embeddings are similarly initialized from a KG embedding model or LLM encoder. Additionally, the question embedding $\mathbf{q} \in \mathbb{R}^d$ is obtained from a separate BERT encoder model (Devlin et al., 2019) that processes the input question. Our GNN updates the node and edge representations iteratively using a multi-layer message-passing scheme. At each layer $l$, we first update the edge embeddings by aggregating information from their neighboring

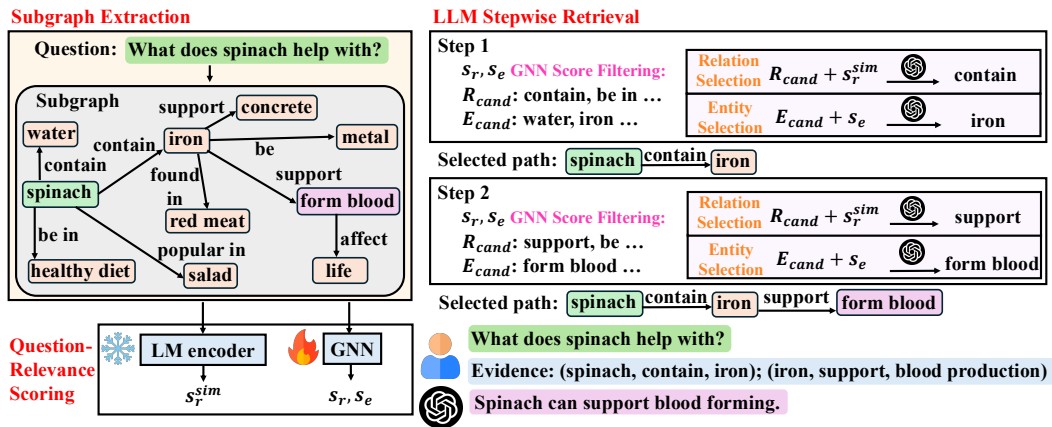

Figure 2: Overview of our proposed framework GGR. The framework consists of: (1) Subgraph Extraction, where contextualized subgraphs are extracted from the KG based on entities mentioned in the question; (2) Question-Relevance Scoring, where we calculate question-relevance scores based on a trainable graph neural network and a fixed LM encoder; (3) LLM Stepwise Retrieval, where relation and entity candidates are filtered from the GNN scores and the LLM retrieves relevant knowledge paths guided by GNN scores and semantic similarity scores; the selected knowledge triplets are then used to construct prompts. $R_{cand}, E_{cand}$ are the retrieval candidates of relations and entities. Finally, the constructed prompts are fed into the LLM alongside the question to generate the answer.

nodes. Given an edge $(h, r, t)$ connecting entities $h$ and $t$ through relation $r$, the updated edge representation is computed as:

$$\mathbf{h}_r^{(l)} = f_{\text{edge}}^{(l)} \left( \mathbf{h}_h^{(l-1)} || \mathbf{h}_t^{(l-1)} || \mathbf{h}_r^{(l-1)} || \mathbf{q} \right), \tag{1}$$

where $\mathbf{h}_h^{(l-1)}, \mathbf{h}_t^{(l-1)}$ are the representations of the head and tail entities from the previous layer, $\mathbf{h}_r^{(l-1)}$ is the relation embedding, and $\mathbf{q}$ is the question embedding. The function $f_{\text{edge}}^{(l)}$ is an MLP that processes these inputs to update the edge representation. Next, we update the entity representations by aggregating information from neighboring edges. For each entity $e \in \mathcal{E}_q$, its updated representation is computed as:

$$\mathbf{A}_e = \sum_{(e, r, e') \in \mathcal{T}_q} \mathbf{h}_r^{(l)} + \sum_{(e', r, e) \in \mathcal{T}_q} \mathbf{h}_r^{(l)}, \qquad \mathbf{h}_e^{(l)} = f_{\text{node}}^{(l)} \left( \mathbf{h}_e^{(l-1)} || \mathbf{A}_e \right), \tag{2}$$

where $\mathbf{A}_e$ represents the sum of messages from all edges to $e$, and $f_{\text{node}}^{(l)}$ is an MLP layer that combines the entity embeddings at previous layers with the aggregated edge embeddings. After $L$ layers of message passing, we compute Question-Relevance scores for each entity and relation using separate scoring networks:

$$s_e = \sigma(F_{\text{node}}(\mathbf{h}_e^{(L)})), \quad s_r = \sigma(F_{\text{edge}}(\mathbf{h}_r^{(L)})), \tag{3}$$

where $s_e$ and $s_r$ represent the final scores for entities and relations, $\sigma(\cdot)$ is the sigmoid function ensuring outputs are in $(0, 1)$, and $F_{\text{node}}$ and $F_{\text{edge}}$ are MLPs for nodes and edges.

These scores play a crucial role in the retrieval phase by providing a global view of the importance of entity and relation. Unlike traditional stepwise LLM-based selection, which considers only local KG knowledge, our GNN scoring ensures that important multi-hop reasoning connections are preserved via incorporating global information.

We emphasize that the GNN in our framework is designed to be lightweight and modular. Its purpose is not to perform deep or complex reasoning, but to provide soft, interpretable relevance signals that assist the LLM during step-wise retrieval. Since the final reasoning and answer generation are handled by the LLM, a highly expressive or deep GNN is not necessary, and our framework remains effective even with simple architectures.

**Path-Supervised GNN Optimization.** Next, we will describe how the GNN mentioned above is optimized. To enable effective knowledge selection, the GNN is optimized to assign meaningful relevance scores to entities and relations within the retrieved contextualized subgraph in a Path Supervision Training step. Given a question $q$ and its corresponding answer $a$, we first identify the set of entities $\mathcal{E}_q$ appearing in the question and the set of entities $\mathcal{E}_a$ appearing in the ground truth answer. To construct training supervision, we search for paths in the knowledge graph that connect any entity in $\mathcal{E}_q$ to any entity in $\mathcal{E}_a$ within a predefined hop limit. Let $\mathcal{P}_q$ denote the set of such paths. Entities and relations that appear on any path in $\mathcal{P}_q$ are treated as positive samples, denoted as as $\mathcal{E}_q^+$ and as $\mathcal{R}_q^+$, while all other entities and relations in the contextualized subgraph are treated as negative samples, denoted as as $\mathcal{E}_q^-$ and as $\mathcal{R}_q^-$.

The GNN is trained to assign high scores to positive samples and low scores to negative samples using the loss $\mathcal{L}$ containing the loss for entities and relations:

$$\mathcal{L}_{\text{entity}} = -\frac{1}{|\mathcal{E}_q^+|} \sum_{e \in \mathcal{E}_q^+} \log(s_e) - \frac{1}{|\mathcal{E}_q^-|} \sum_{e \in \mathcal{E}_q^-} \log(1 - s_e), \tag{4}$$

$$\mathcal{L}_{\text{relation}} = -\frac{1}{|\mathcal{R}_q^+|} \sum_{r \in \mathcal{R}_q^+} \log(s_r) - \frac{1}{|\mathcal{R}_q^-|} \sum_{e \in \mathcal{R}_q^-} \log(1 - s_r), \tag{5}$$

$$\mathcal{L} = \mathcal{L}_{\text{entity}} + \mathcal{L}_{\text{relation}}, \tag{6}$$

where $s_e$ and $s_r$ are the predicted scores for entities and relations.

During training, the GNN learns to refine entity and relation relevance scores through iterative message passing, guided by path-based supervision. The optimization objective ensures that entities and relations contributing to valid reasoning paths receive higher scores, while irrelevant ones are suppressed. By minimizing $\mathcal{L}$, the model progressively enhances its ability to capture multi-hop dependencies and structural importance within the KG, providing a stronger retrieval signal for downstream reasoning.

**Semantic Similarity Scores.** We compute a query-relation semantic similarity score for each relation $r$: $s_r^{\text{sim}} = \cos(\mathbf{q}, \mathbf{e}_r)$, where $\mathbf{q}$ and $\mathbf{e}_r$ are respectively the embeddings of the question and the relation text obtained from an LM encoder, typically a BERT encoder model (Devlin et al., 2019). We calculate the cosine similarity between these embeddings as the semantic similarity score. These similarity scores are used to refine the question-relevance information of relations.

### 3.3 LLM Stepwise Retrieval

After obtaining the question-relevance scores for entities and relations, we perform knowledge retrieval by LLMs to extract relevant knowledge for LLM reasoning with the assistance of the scores. Our method follows a stepwise retrieval strategy starting from the entities appearing in the question. At each hop, we first select relations and then select entities based on the chosen relations.

In this phase, the predicted GNN scores are treated as auxiliary information for the LLM to reason so we can include the global KG information while retaining the ability of LLM reasoning in the retrieval. However, though the GNN provides relevance scores based on global KG information, its training supervision is derived solely from path connectivity, which may include intermediate nodes and relations that are structurally valid but semantically irrelevant to the query. As a result, relying exclusively on these scores for retrieval may lead to the inclusion of irrelevant or noisy knowledge. To address this, we propose to incorporate the local semantic similarity scores into the retrieval process. This refinement step complements the GNN's global perspective with semantic signals, enabling more accurate and context-aware knowledge selection.

We denote the set of question-relevant entities selected in retrieval step $t$ as $\mathcal{E}_q^{(t)}$. Let the initial entity set be $\mathcal{E}_q^{(0)} = \mathcal{E}_q$, and initialize the retrieved triplet set as $\mathcal{T}_q' = \emptyset$. At each step

$t$, the retrieval proceeds in two stages: relation selection and entity selection. To enable flexible control over retrieval decisions, we group all the candidates into three categories based on their GNN scores $s_x$ ($x$ can be an entity or a relation) using thresholds $\tau_h$ and $\tau_l$, which enables **GNN Score Filtering** in the retrieval step:

$$\mathcal{X}_{\text{high}} = \{x \mid s_x \geq \tau_h\}, \quad \mathcal{X}_{\text{low}} = \{x \mid s_x \leq \tau_l\}, \quad \mathcal{X}_{\text{mid}} = \{x \mid \tau_l < s_x < \tau_h\}. \tag{7}$$

High-scoring candidates ($\mathcal{X}_{\text{high}}$) are directly selected, while low-scoring ones ($\mathcal{X}_{\text{low}}$) are discarded. Mid-scoring candidates ($\mathcal{X}_{\text{mid}}$) are regarded as relation or entity candidates for LLM selection. We adopt a selection strategy by including candidates and their scores together in a prompt and use the LLM to reason over their relevance, as shown in Appendix E. This approach avoids the pruning of relevant knowledge using global graph information.

Specifically, at each retrieval step $t$, we expand from the entity set $\mathcal{E}_q^{(t-1)}$ by examining their neighboring relations and entities. For an entity $e' \in \mathcal{E}_q^{(t-1)}$, we first identify all triplets $T = (e', r, e)$ or $(e, r, e')$ that haven't been explored in the subgraph $\mathcal{G}_q$. We directly select $r \in \mathcal{X}_{\text{high}}$ and discard $r \in \mathcal{X}_{\text{low}}$. For $r \in \mathcal{X}_{\text{mid}}$, we use the query-relation semantic similarity $s_r^{\text{sim}}$ and pass both $r$ and $s_r^{\text{sim}}$ into the LLM prompt. The LLM then determines which relations to retain based on their relevance to the question. By incorporating $s_r^{\text{sim}}$, we ensure that relations that may have a lower GNN score but are semantically aligned with the query are not overlooked. This step complements the GNN's structural scoring.

For each retained relation $r$, all the neighboring entity $e$ is evaluated using its GNN score $s_e$. Similar to relation selection, entities with high scores are directly selected, low scores are discarded, and mid-scoring entities are passed into the LLM along with their GNN scores for question-relevance judgment because entity relevance often depends more on contextual reasoning over the graph structure. If the LLM selects the entity, we add the corresponding triplet $T$ to the retrieved triplet set $\mathcal{T}_q'$ and expand the entity set $\mathcal{E}_q^{(t)}$ initializing as $\varnothing$. These sets are updated as follows:

$$\mathcal{T}_q' \leftarrow \mathcal{T}_q' \cup \{T\}, \quad \mathcal{E}_q^{(t)} \leftarrow \mathcal{E}_q^{(t)} \cup \{e\}. \tag{8}$$

This process repeats for a fixed number of steps or until no new entities are added. The final set $\mathcal{T}_q'$ is converted into natural language and used to construct the input prompt for the LLM. By combining global relevance scores from the GNN with query-relation semantic similarity and deferring fine-grained reasoning to the LLM, this retrieval strategy achieves flexible and precise knowledge selection. It mitigates retrieval errors by maintaining plausible candidates and using semantic alignment.

### 3.4 Prompt Construction and LLM Reasoning

With the selected set of triplets $\mathcal{T}_q'$ obtained from the retrieval step, we construct the prompt for the LLM. Specifically, we convert all triplets $(h, r, t) \in \mathcal{T}_q'$ into a textual input for the LLM. The final LLM input consists of the original question $q$ and the extracted knowledge triplets $\mathcal{T}_q'$. The generated answer $a_q$ given by the LLM is represented as $a_q = LM(q, \mathcal{T}_q')$. Given the prompt, the LLM generates an answer by leveraging both its pertrained knowledge and the external KG evidence. The integration of structured knowledge ensures that the response is grounded in factual information while allowing for more informed reasoning based on the retrieved evidence. By incorporating both global importance from the GNN and local semantic alignment through retrieval, this approach enhances the accuracy and reliability of the generated answers compared to conventional KG-RAG methods.

## 4 Experiments

In this section, we evaluate the effectiveness of our proposed approach through multiple experiments. We compare our method against multiple baselines to assess its ability to retrieve and utilize knowledge during KG-RAG by LLMs to improve the performance of LLM-based question answering.

### 4.1 Datasets

We evaluate our method on WebQuestionsSP (WebQSP) (Yih et al., 2016) and ComplexWe-bQuestions (CWQ) (Talmor & Berant, 2018), two benchmark datasets designed for question answering over knowledge graphs. For both datasets, we use Freebase (Bollacker et al., 2008), a large-scale knowledge graph in diverse domains, as the external knowledge base. WebQSP consists of 4,737 natural language questions, each paired with SPARQL queries and answer entities, making it suitable for assessing structured reasoning over a KG. CWQ introduces more complex, multi-hop questions that require reasoning over multiple triplets. It contains 34,689 questions, requiring compositional and logical reasoning over Freebase. These datasets provide a comprehensive evaluation ground for measuring the effectiveness of retrieval and reasoning in knowledge-enhanced LLM-based question answering.

### 4.2 Baselines

To evaluate the effectiveness of our proposed method, we compare it against several baseline approaches that represent different retrieval and reasoning strategies for knowledge-enhanced question answering. (1) **IO-prompt** (Brown et al., 2020): This baseline uses the LLMs to answer the questions without any additional enhancements. (2) **CoT-prompt** (Wei et al., 2022): Chain-of-Thought (CoT) prompting instructs the LLM to generate intermediate reasoning steps before producing a final answer. By explicitly modeling the reasoning process, this method improves the model's ability to answer multi-step questions. (3) **Self-Consistency** (Wang et al., 2022): This approach extends CoT prompting by sampling multiple reasoning paths and selecting the most consistent answer among them. By aggregating diverse reasoning trajectories, it reduces response variability. (4) **Sim-Retrieve** (Baek et al., 2023a): In this retrieval-based baseline, we utilize an LLM encoder to compute the semantic similarity between the input question and candidate knowledge triplets. The top-ranked triplets based on similarity scores are retrieved and incorporated into the prompt for answer generation. (5) **GNN-Scoring**: This baseline uses the GNN-trained relevance scores to directly retrieve knowledge triplets from the KG. Instead of relying on an LLM-based measure, it selects the highest-scoring triplets by GNNs as evidence for question answering. (6)**Think-on-Graph (ToG)** (Sun et al., 2024): Think-on-Graph integrates structured knowledge from a knowledge graph into the LLM's reasoning process, representing the methods using LLM generation to perform knowledge retrieval. It utilizes KG-based retrieval to enrich the model's input, allowing it to leverage explicit entity-relation structures for improved factual accuracy. At each step, it uses the LLM to decide whether to continue retrieving or terminate the process.

### 4.3 Experimental Settings

To assess the effectiveness of our method, we conduct experiments using **GPT-3.5** (OpenAI, 2022), **GPT-4o-mini** (Anand et al., 2023), and **Claude-3-haiku** (Anthropic, 2024) as the LLMs. The selection process is controlled by two threshold values: a high-confidence threshold $\tau_h = 0.9$ and a low-confidence threshold $\tau_l = 0.1$, which regulate the filtering of retrieved entities and relations. All experiments are conducted five times, and we report the **average results** to reduce variance. More implementation details are shown in Appendix B.

### 4.4 Results and Analysis

From the experimental results presented in Table 1, we can observe that our method consistently outperforms all baselines, demonstrating the effectiveness of integrating GNN-based scoring and query-relation semantic alignment for knowledge retrieval and QA performance. Among the baselines, ToG achieves the best results, highlighting the advantages of leveraging LLMs for retrieval on structured knowledge graphs. However, its reliance on LLM-only knowledge extraction leads to errors, particularly when important reasoning paths are discarded due to the lack of KG knowledge in a broader view. Our GNN scoring propagates question-relevance information across the graph, preserving globally important knowledge paths that LLM retrieval methods often miss. Our method also incorporates query-relation semantic similarity, ensuring that semantically relevant relations are correctly retained. Sim-Retrieve and GNN-Scoring exhibit inconsistent performance, performing

Table 1: Experimental results (accuracy in %) of GGR and all baselines on two QA datasets. The best and second-best results are shown in **bold** and underlined, respectively.

| Method | GPT-3.5 | | GPT-4o-mini | | Claude-3-haiku | |
|---|---|---|---|---|---|---|
| | **WebQSP** | **CWQ** | **WebQSP** | **CWQ** | **WebQSP** | **CWQ** |
| **IO-prompt** | $62.32 \pm 0.08$ | $36.83 \pm 0.10$ | $63.77 \pm 0.07$ | $38.99 \pm 0.07$ | $74.01 \pm 0.09$ | $33.30 \pm 0.09$ |
| **CoT-prompt** | $62.74 \pm 0.21$ | $38.45 \pm 0.27$ | $64.56 \pm 0.24$ | $39.43 \pm 0.22$ | $76.78 \pm 0.26$ | $36.83 \pm 0.25$ |
| **Self-Consistency** | $61.11 \pm 0.06$ | $46.97 \pm 0.09$ | $61.90 \pm 0.08$ | $47.76 \pm 0.08$ | $70.19 \pm 0.10$ | $43.52 \pm 0.07$ |
| **Sim-Retrieve** | $46.98 \pm 0.63$ | $32.75 \pm 0.66$ | $48.94 \pm 0.57$ | $32.17 \pm 0.60$ | $29.75 \pm 0.70$ | $20.81 \pm 0.68$ |
| **GNN-Scoring** | $63.86 \pm 0.19$ | $38.33 \pm 0.24$ | $65.90 \pm 0.23$ | $38.19 \pm 0.20$ | $67.48 \pm 0.21$ | $35.22 \pm 0.22$ |
| **Think-on-Graph** | $\underline{74.24} \pm 0.55$ | $\underline{55.92} \pm 0.60$ | $\underline{80.32} \pm 0.61$ | $\underline{66.41} \pm 0.58$ | $\underline{83.24} \pm 0.52$ | $\underline{60.16} \pm 0.56$ |
| **GGR** | $\mathbf{83.27} \pm 0.60$ | $\mathbf{63.48} \pm 0.59$ | $\mathbf{90.50} \pm 0.67$ | $\mathbf{70.33} \pm 0.65$ | $\mathbf{91.89} \pm 0.62$ | $\mathbf{65.35} \pm 0.61$ |

even worse than knowledge-free methods like IO-prompt in some experimental settings. This is due to the retrieval noise introduced by selecting suboptimal triplets. Sim-Retrieve overemphasizes embedding similarity without considering structural relevance, while GNN-Scoring lacks LLM reasoning. In contrast, our method balances global subgraph structure and local semantic similarity, preventing irrelevant knowledge from misleading the LLM. Furthermore, since our GNN is small, the overall inference time remains comparable to standard LLM-based KG-RAG methods. In summary, by integrating GNN graph reasoning with semantic embedding information in LLM reasoning process, our approach enables more accurate and context-aware retrieval, leading to more reliable LLM responses and significantly improved KG-RAG performance.

To further examine the role of the GNN component, we replaced our default lightweight GNN with a more complex architecture used in GNN-RAG (Mavromatis & Karypis, 2024), which includes attention-based message passing. As shown in Table 2, this study yields moderate improvements across all LLMs and datasets. These results indicate that while a stronger GNN can enhance retrieval quality slightly, the overall performance gains are limited. A simple GNN is sufficient for guiding retrieval with relevance signals.

Table 2: Experimental results (accuracy in %) of simple and complex GNN in GGR on two QA datasets.

| LLM | GNN | WebQSP | CWQ |
|---|---|---|---|
| **GPT-3.5** | **Simple** | 83.27 | 63.48 |
| | **Complex** | 85.72 | 66.25 |
| **GPT-4o-mini** | **Simple** | 90.50 | 70.33 |
| | **Complex** | 91.72 | 73.39 |
| **Claude-3-haiku** | **Simple** | 91.89 | 65.35 |
| | **Complex** | 93.53 | 67.66 |

## 4.5 Parameter Study

Our method utilizes two thresholds, $\tau_h$ and $\tau_l$, to control how relation and entity candidates are categorized during the retrieval process. This subsection investigates how variations in these two parameters affect final QA performance.

We conduct experiments on the WebQSP dataset to evaluate the effect of the two thresholds $\tau_h$ and $\tau_l$. Using GPT-3.5 as the LLM, we evaluate all combinations of $\tau_h \in \{0.70, 0.80, 0.90, 1.00\}$ and $\tau_l \in \{0.00, 0.05, 0.10, 0.15\}$. The results are shown in Figure 3.

From the results, we can observe that setting low $\tau_h$ results in many low-confidence candidates being directly included, increasing noise in the prompt and leading to degraded performance. On the other hand,

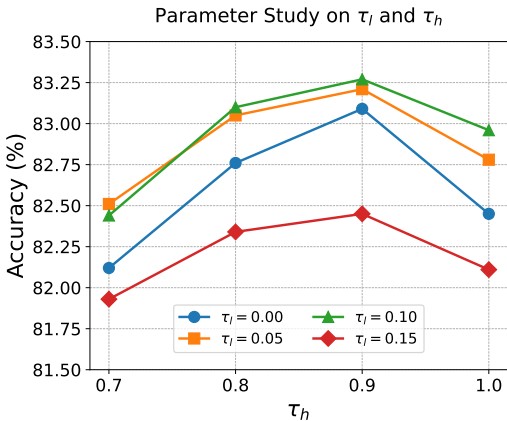

Figure 3: Parameter sensitivity results on dataset WebQSP using GPT-3.5. We vary the values of $\tau_h$ and $\tau_l$ to examine the effect of retrieval thresholds in terms of QA accuracy.

Table 3: Experimental results of different variants of our method on two datasets (accuracy in %). The best results are shown in **bold**.

| Method | GPT-3.5 | | GPT-4o-mini | | Claude-3-haiku | |
|---|---|---|---|---|---|---|
| | WebQSP | CWQ | WebQSP | CWQ | WebQSP | CWQ |
| GGR | **83.27** | **63.48** | **90.50** | **70.33** | **91.89** | **65.35** |
| GGR w/o GScore | 78.62 | 59.48 | 84.29 | 67.94 | 85.05 | 62.05 |
| GGR w/o SSim | 80.47 | 59.96 | 85.72 | 68.41 | 86.20 | 62.39 |
| GGR w/o GScore, SSim | 74.04 | 56.32 | 83.03 | 65.35 | 83.14 | 60.50 |
| GGR w/o QEnc | 80.17 | 60.79 | 84.31 | 66.63 | 84.26 | 61.28 |

overly high $\tau_h$ values make the retrieval process overly conservative, potentially excluding useful knowledge paths. Similarly, setting a high $\tau_l$ causes more mid-scoring candidates to be discarded prematurely, harming retrieval coverage. Conversely, low $\tau_l$ increases the burden on the LLM by passing more uncertain candidates as prompts, which may also slightly hurt performance. These results highlight the importance of balancing different types of reasoning of retrieval.

## 4.6 Ablation Study

To evaluate the impact of each component, we conduct an ablation study by removing key modules in our framework. **w/o GScore** eliminates GNN scoring, preventing global question-relevance information propagation and leading to the premature loss of important reasoning paths, making multi-hop retrieval less reliable. **w/o SSim** removes query-relation semantic similarity scoring, which causes frequent errors in relation selection as the model struggles to retain semantically relevant relations that are predicted to have lower GNN scores. Without this refinement, retrieval relies solely on LLM textual matching and graph structure matching, increasing the chance of missing key knowledge. **w/o GScore, SSim** leads to the most severe performance degradation, as removing both structural reasoning and semantic alignment reduces retrieval to a purely stepwise expansion, making it highly prone to filtering errors and irrelevant triplet selection. **w/o QEnc** removes the question embedding from the GNN, preventing the model from conditioning relevance scores on the query, which results in noisier retrieval and lower precision. From the results in Table 3, we can observe that both scores play crucial roles in balancing global structural reasoning and local semantic refinement, while incorporating question information further enhances retrieval quality by ensuring that selected entities and relations align with the specific query.

## 5 Related Work

**Retrieval-Augmented Generation.** Retrieval-Augmented Generation (RAG) has emerged as a powerful approach to enhance the factual accuracy and reasoning capabilities of LLMs by incorporating external knowledge. Earlier works on RAG primarily focused on retrieving unstructured textual information from large corpora such as Wikipedia or domain-specific databases (Karpukhin et al., 2020; Lewis et al., 2020b). Prevalant retrieval methods include sparse retrieval techniques like BM25 (Robertson & Zaragoza, 2009) and dense retrieval models such as DPR (Karpukhin et al., 2020). These methods often rely on free-text evidence, which lacks explicit relational structure, making them less effective for tasks requiring structured reasoning.

**Knowledge Graphs for Question Answering and KG-RAG.** To address the limitations of unstructured retrieval, KGs have been widely used in question answering (QA) due to their ability to provide structured and semantically rich representations of knowledge (Liu et al., 2019; Xu et al., 2024; Jia et al., 2019; Baek et al., 2023b). Traditional KG-based QA approaches rely on symbolic reasoning methods, such as SPARQL-based queries (Berant et al., 2013), or graph traversal techniques to extract relevant facts (Sun et al., 2018). More recently, KG-RAG methods have been proposed to integrate knowledge graphs into LLM-based retrieval (Yasunaga et al., 2022b). These approaches typically follow a two-step process: (1) retrieving relevant entities and relations based on the query; (2) integrating the retrieved

KG subgraph into the LLM's prompt to guide answer generation. Most existing KG-RAG methods rely on LLMs to evaluate triplet relevance. However, such stepwise expansion merely considers local KG knowledge and leads to errors where important multi-hop reasoning paths are discarded prematurely.

**Graph Neural Networks for Knowledge Graph Reasoning.** Graph Neural Networks (GNNs) have been widely used in knowledge graph tasks such as node classification, link prediction, and entity ranking (Kipf & Welling, 2016; Veličković et al., 2017; Wu et al., 2020). By propagating information across graph structures, GNNs can capture global context and identify important relationships between entities (Hamilton et al., 2017; You et al., 2018; Gao & Ji, 2019; Liu et al., 2022). However, most existing KG-RAG methods do not incorporate graph-based reasoning into the retrieval process, instead relying on LLM-based triplet selection. In contrast, our approach enhances retrieval quality through GNN-guided prompting by integrating GNN-based scoring to provide global structural awareness, ensuring that key reasoning paths are preserved even when their immediate relevance appears weak.

## 6 Conclusion

In this paper, we propose GGR, a GNN-guided KG-RAG framework that enhances knowledge retrieval for large language models (LLMs) by addressing the limitations of existing methods: the reliance on local, stepwise LLM retrieval decisions. To mitigate the issue, GGR leverages a GNN to score entities and relations within a contextualized subgraph, capturing global KG information and preserving reasoning paths that might otherwise be pruned. Additionally, we incorporate query-relation semantic similarity as a signal to refine relation selection, ensuring that semantically relevant knowledge is retained even when predicted to have a lower GNN score. Our framework leverages GNNs purely for scoring and preserves the reasoning capabilities of the LLM, rather than dominating the reasoning process using only GNNs. This design promotes modularity and allows future work to explore combining GGR with stronger GNNs or symbolic engines. Experimental results on knowledge-intensive QA benchmarks demonstrate that GGR consistently improves retrieval quality and generation performance, showcasing the value of integrating graph-based reasoning with LLM reasoning. Future work includes extending GGR to more complex multi-hop reasoning tasks and further exploring adaptive mechanisms between graph retrieval and LLM reasoning.

## 7 Acknowledgement

This work is supported in part by the National Science Foundation (NSF) under grants IIS-2006844, IIS-2144209, IIS-2223769, CNS-2154962, BCS-2228534, and CMMI-2411248; the Office of Naval Research (ONR) under grant N000142412636; and the Commonwealth Cyber Initiative (CCI) under grant VV-1Q24-011.

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

## A Limitations

While our proposed GGR framework demonstrates strong performance in KG-RAG tasks by integrating global structural scoring and local semantic alignment, it still has several limitations. First, the effectiveness of the GNN-based scoring relies on the quality of the retrieved subgraph. If key entities or paths are missing in the original graph, the GNN may be unable to assign meaningful scores, limiting its contribution to retrieval. Second, although the semantic similarity scoring helps address phrasing mismatches between queries and relations, it is limited by the expressiveness of the underlying embedding model (e.g., DistilBERT), which may not capture deeper contextual nuances. Third, the stepwise retrieval process requires multiple calls to the LLM, which could impact efficiency and latency in real-time applications. We leave these challenges as directions for future work.

## B Experimental Settings

We conduct experiments on two widely-used knowledge-based question answering benchmarks: WebQuestionsSP (WebQSP) (Yih et al., 2016) and ComplexWebQuestions (CWQ) (Talmor & Berant, 2018), both grounded in the Freebase knowledge graph (Bollacker et al., 2008). These datasets are selected to evaluate multi-hop reasoning capabilities of methods over structured knowledge.

To evaluate the generality of our method across different LLMs, we apply our approach to three models with varying capacities: GPT-3.5, GPT-4o-mini, and Claude-3-haiku. All evaluations are conducted in a zero-shot setting using a unified prompt template to ensure fair comparison across models.

Our retrieval module is based on a three-layer graph neural network (GNN) with a hidden dimension of 2,048. The input to the GNN includes both node and relation embeddings, which are initialized using a pretrained DistilBERT encoder. Specifically, relation and entity names are passed through the encoder, and the resulting 768-dimensional embeddings are further projected to 128 dimensions before being input into the GNN. Question embeddings used during scoring are also obtained from the same DistilBERT encoder, also projected to 128 dimensions before being input into the GNN. When calculating the semantic similarity scores, the embeddings remain in 768 dimensions.

During GNN training, path-based supervision is used with a maximum path length of 3 hops between question and answer entities. For filtering candidates during retrieval, we adopt a flexiable selection strategy governed by two thresholds: a high-confidence threshold $\tau_h = 0.9$ and a low-confidence threshold $\tau_l = 0.1$. These thresholds are used to categorize candidate entities and relations into high-, mid-, and low-confidence groups, which inform the retrieval decision process and the construction of prompts. The retrieval steps for WebQSP and CWQ are 2 and 3, respectively. The maximum number of candidates being selected are set to 3.

All experiments are run five times with different random seeds, and we report the mean accuracy and standard deviation to account for variability. Our implementation is based on PyTorch and Python 3.11.7, and all experiments are conducted on NVIDIA A100 GPUs with 80GB memory.

We release our code at `https://github.com/HaochenLiu2000/GGR`.

## C Datasets

Our experiments are conducted on three popular knowledge graph question answering (KGQA) datasets: WebQuestionsSP (WebQSP) (Yih et al., 2016), ComplexWebQuestions (CWQ) (Talmor & Berant, 2018). Table 4 summarizes the statistics for each dataset.

WebQSP features 4,737 natural language questions, each answerable via a subset of the Freebase knowledge graph (Bollacker et al., 2008), which contains 24.9 million entities and approximately 164.6 million triples. Answering these questions often involves multi-hop

Table 4: The statistics of the datasets used in our experiments. "Sub. Size" denotes the average number of entities in the subgraph, and "Coverage" denotes the percentage of subgraphs containing at least one answer.

| Datasets | Train | Dev | Test | Sub. Size | Coverage (%) |
|---|---|---|---|---|---|
| WebQSP | 2,848 | 250 | 1,639 | 1,429.8 | 94.9 |
| CWQ | 27,639 | 3,519 | 3,531 | 1,305.8 | 79.3 |

reasoning: around 30% require reasoning across two facts, 7% demand constraint-based inference, and the remaining can be answered using a single fact from the KG.

CWQ builds upon WebQSP by modifying the original questions to increase complexity, such as extending the scope of entities or introducing additional constraints. The dataset includes 34,689 questions, categorized into four types: composition (45%), conjunction (45%), comparative (5%), and superlative (5%). These questions require reasoning over paths of up to four hops, still within the Freebase KG.

## D   Large Language Models

- **GPT-3.5** (OpenAI, 2022) is a widely used large language model developed by OpenAI. It builds upon the success of GPT-3 by incorporating architectural improvements and fine-tuning techniques to enhance reasoning, code generation, and few-shot learning performance. GPT-3.5 is optimized for general-purpose tasks and serves as a strong baseline for evaluating retrieval-augmented generation systems.

- **GPT-4o-mini** (Anand et al., 2023) is a smaller variant of OpenAI's GPT-4o series, designed to offer a balance between capability and computational efficiency. Despite its compact size, GPT-4o-mini maintains competitive performance across a range of NLP tasks, including question answering and reasoning. Its lightweight nature makes it particularly suitable for exploring the scalability of knowledge-enhanced language models.

- **Claude-3-haiku** (Anthropic, 2024) is part of the Claude-3 model family introduced by Anthropic. Haiku is the smallest and most efficient model in the Claude-3 lineup, optimized for low-latency applications while maintaining robust performance in multi-turn dialogue and knowledge-intensive tasks. Its design emphasizes safety and alignment, making it an increasingly popular choice for controlled QA applications.

## E   Prompts

In this subsection, we list the prompts we use for Relation Selection, Entity Selection and the final Question Answering. When using them, we replace the contents in {} with our target data.

---

**Question Answering Prompt**

Based on the knowledge triplets, please answer the given question. Please keep the answer as simple as possible and list all your answers.
Question: {Query}
Knowledge Triplets: {Evidence Text}
Answer: 1. ...

---

**Relation Selection Prompt**

Please retrieve relations (separated by semicolon) that contribute to the question and you can use the score that is a scale from 0 to 1 as a reference. You still need to identify the relations yourself. You can select at most {candidate number} candidates.
Question: Name the president of the country whose main spoken language was Brahui in 1980?
Topic Entity: Brahui Language
Relations:
language.human_language.main_country(0.8);
language.human_language.language_family(0.3);
language.human_language.iso_639_3_code(0.3);
base.rosetta.languoid.parent(0.6);
language.human_language.writing_system(0.3);
base.rosetta.languoid.languoid_class(0.2);
language.human_language.countries_spoken_in(0.8);
kg.object_profile.prominent_type(0.3);
base.rosetta.languoid.document(0.1);
base.ontologies.ontology_instance.equivalent_instances(0.3);
base.rosetta.languoid.local_name(0.3);
language.human_language.region(0.3);
Answer:
1. {language.human_language.main_country}: This relation is highly relevant as it directly relates to the country whose president is being asked for, and the main country where Brahui language is spoken in 1980.
2. {language.human_language.countries_spoken_in}: This relation is also relevant as it provides information on the countries where Brahui language is spoken, which could help narrow down the search for the president.
3. {base.rosetta.languoid.parent}: This relation is less relevant but still provides some context on the language family to which Brahui belongs, which could be useful in understanding the linguistic and cultural background of the country in question.

Question: {Query}
Topic Entity: {Entity}
Relations: {Relations}
Answer: 1. ...

---

**Entity Selection Prompt**

Please retrieve the entities that contribute to the question and you can use the score given by graph models that is a scale from 0 to 1 as a reference. You still need to identify the entities yourself. You can select at most {candidate number} candidates.
Question: The movie featured Miley Cyrus and was produced by Tobin Armbrust?
Relation:
film.producer.film
Entites:
The Resident(0);
So Undercover(1);
Let Me In(0);
Begin Again(0);
The Quiet Ones(0);
A Walk Among the Tombstones(0);
Answer:
1. {So Undercover}: The movie that matches the given criteria is "So Undercover" with Miley Cyrus and produced by Tobin Armbrust.
2. {Let Me In}: The movie "Let Me In" is produced by Tobin Armbrust.

Question: {Query}
Relation: {Relation}
Entities: {Entities}
Answer: 1. ...

---

## F   Ethics Statement

This work focuses on improving knowledge retrieval for question answering using publicly available language models and knowledge graphs. All experiments are conducted on benchmark datasets without involving any personally identifiable or sensitive information. We do not perform any fine-tuning of proprietary LLMs. While our method enhances the factual grounding of Question-Answering systems, care should be taken when applying it to domains with high-stakes decisions, such as healthcare or law, to avoid over-reliance on model-generated outputs.

