# OpenReview forum: "Knowledge Graph Retrieval-Augmented Generation via GNN-Guided Prompting"
_colmweb.org/COLM/2025/Conference — COLM 2025_

### Official Review · Reviewer_w5S6 · 2025-05-09

**Rating:** 6
**Confidence:** 5
**Ethics Flag:** 1

**Summary:**

The paper investigates Knowledge Graph Retrieval-Augmented Generation (KG-RAG) for Knowledge Graph Question Answering (KGQA) tasks. It addresses the limitation of Large Language Model (LLM) approaches, which are constrained by their ability to process only local context from knowledge subgraphs. To overcome this, the authors introduce Graph-Contextualized Retrieval (GCR), a methodology that employs Graph Neural Networks (GNNs) specifically to model long-range dependencies between entities and relations within knowledge subgraphs. The GNN component is trained using path-based supervision to score the relevance of entities and relations with respect to a given question. During inference, GCR implements a hybrid scoring mechanism that combines LLM-based filtering, GNN scoring, and semantic scoring to guide the LLM toward identifying correct answer entities. Empirical evaluations conducted on the WebQSP and CWQ) datasets demonstrate that GCR achieves performance improvements over previous LLM-based traversal methods such as ToG.

**Questions To Authors:**

- How many layers does the proposed GNN consist of? By looking at the code, it seems you use num_layers=3, however there are no ablations provided to verify the claim that the GNN captures more global graph information (e.g., 1-layer GNN cannot capture this). I would recommend adding ablation studies on the GNN number of layer hyperparameter.

- Could we replace the proposed GNN-scoring with established KGQA-GNNs to further improve the performance in KGQA?

**Reasons To Accept:**

- The paper is clearly written and well-motivated
- Experimental results demonstrate that the step-wise hybrid scoring approach (combining GNN scoring, semantic scoring, and LLM-based filtering) outperforms LLM-based traversal methods like ToG across various backbone LLMs.
- The proposed path-based supervision represents a more advanced approach to GNN training compared to previous methods that rely solely on QA pairs, although it should be noted that path-based supervision is already established in KGQA literature (e.g., in approaches like RoG [1]).

[1] Luo, Linhao, et al. "Reasoning on graphs: Faithful and interpretable large language model reasoning." arXiv preprint arXiv:2310.01061 (2023).

**Reasons To Reject:**

- The novelty of the paper is limited within the field of KGQA. The authors omit discussion and comparison with many relevant works (e.g., RoG [1]), particularly GNN-based approaches that have been developed since 2018 ([2,3]). GNN-based approaches have been widely adopted in KGQA tasks and recently extended within RAG-based inference frameworks (GNN-RAG [4]). However, the paper does not compare against these approaches or elaborate on how GCR captures information that previous GNN models cannot. I recommend expanding both the experimental section and related work sections to include these important comparisons.

- The proposed GNN scoring appears to underperform on KGQA tasks (as shown in Table 1, where GNN scoring underperforms compared to LLMs alone), raising concerns about whether the GNN component is truly the main driver of improvements. I speculate that the step-wise LLM reasoning might be more significant, and this approach could potentially be extended to work with existing frameworks in the literature (such as NSM, GNN-RAG). For example, when comparing with GNN performance numbers reported in NSM [3] (Table 2), those GNN implementations achieve better performance than the proposed GNN-Score. This suggests the possibility of directly combining those more powerful GNNs with the step-wise LLM reasoning approach presented here.

[1] Luo, Linhao, et al. "Reasoning on graphs: Faithful and interpretable large language model reasoning." arXiv preprint arXiv:2310.01061 (2023).

[2] Sun, Haitian, et al. "Open domain question answering using early fusion of knowledge bases and text." arXiv preprint arXiv:1809.00782 (2018).

[3] He, Gaole, et al. "Improving multi-hop knowledge base question answering by learning intermediate supervision signals." Proceedings of the 14th ACM international conference on web search and data mining. 2021.

[4] Mavromatis, Costas, and George Karypis. "Gnn-rag: Graph neural retrieval for large language model reasoning." arXiv preprint arXiv:2405.20139 (2024).

---

> ### Author Response · Authors · 2025-05-30
> **Response 1/2**
>
> Thank you for your valuable feedback and for highlighting both the strengths and weaknesses of our work. Your comments have been instrumental in guiding us to refine our approach and clarify key points in the paper. We have provided detailed responses below to address each of your concerns and ensure the clarity and robustness of our contributions.
>
> >The authors omit discussion and comparison with many relevant works (e.g., RoG [1])
>
> We thank you for pointing out RoG as an important related work. In our experimental setup, we included Think-on-Graph (ToG), which shares a similar design philosophy with RoG by relying on LLMs to make retrieval decisions over KG paths without external scoring mechanisms. We chose ToG as a competitive representative of LLM-only retrieval-based reasoning frameworks, and GGR consistently outperforms ToG across all evaluated datasets and LLM variants. Additionally, we have reviewed the results reported in the RoG paper and find that GGR also achieves overall higher accuracy. This further supports the effectiveness of integrating GNN-based global scoring into the retrieval loop. While RoG explores task decomposition and relation prediction strategies, our work focuses on enhancing retrieval quality via graph-aware soft relevance signals. We will revise the paper to explicitly discuss RoG and clarify the methodological differences and performance comparisons.
>
> >GNN-based approaches have been widely adopted in KGQA tasks and recently extended within RAG-based inference frameworks (GNN-RAG [4]). However, the paper does not compare against these approaches or elaborate on how GCR captures information that previous GNN models cannot. I recommend expanding both the experimental section and related work sections to include these important comparisons. I speculate that the step-wise LLM reasoning might be more significant, and this approach could potentially be extended to work with existing frameworks in the literature (such as NSM, GNN-RAG). For example, when comparing with GNN performance numbers reported in NSM [3] (Table 2), those GNN implementations achieve better performance than the proposed GNN-Score. This suggests the possibility of directly combining those more powerful GNNs with the step-wise LLM reasoning approach presented here. Could we replace the proposed GNN-scoring with established KGQA-GNNs to further improve the performance in KGQA?
>
> We thank you for this valuable suggestion. GGR is fundamentally different from traditional GNN-based KGQA models such as GNN-RAG, where the GNN is the main reasoning engine used for direct answer prediction or symbolic path traversal. In contrast, GGR treats the GNN as an auxiliary component. Its sole purpose is to assign soft, question-aware relevance scores to nodes and relations within the subgraph. These scores guide the selection of candidate triplets for step-wise reasoning, but do not determine the final answer. The actual reasoning and decision-making are entirely handled by the LLM, which integrates both retrieved facts and question semantics. This design makes the GNN a lightweight and interpretable guide, not a standalone predictor, and enables flexible integration with different LLMs without modifying their architecture or training. In response to your suggestion, we experimented with a stronger GNN architecture given by GNN-RAG, which includes better attention modules. While this variant did improve retrieval quality and downstream QA performance, the gains were moderate. This confirms that although a stronger GNN can offer additional benefits, its role in GGR remains supportive. It helps identify plausible paths, but the LLM is responsible for evaluating and composing reasoning chains. We will expand the experimental section to include these results and clarify the auxiliary role of the GNN throughout the paper.
>
> |LLM| Method                    | WebQSP | CWQ   |
> |-------|---------------------------|--------|-------|
> |GPT-3.5| GGR (ours, simple GNN)    | 83.27  | 63.48 |
> |GPT-3.5| GGR w/ GNN-RAG GNN        | 85.72  | 66.25 |
> |GPT-4o-mini| GGR (ours, simple GNN)    | 90.50  | 70.33 |
> |GPT-4o-mini| GGR w/ GNN-RAG GNN        | 91.72  | 73.39 |
> |Claude-3-haiku| GGR (ours, simple GNN)    | 91.89  | 65.35 |
> |Claude-3-haiku| GGR w/ GNN-RAG GNN        | 93.53  | 67.66 |

---

> > ### Author Response · Authors · 2025-05-30
> > **Response 2/2**
> >
> > >The proposed GNN scoring appears to underperform on KGQA tasks (as shown in Table 1, where GNN scoring underperforms compared to LLMs alone), raising concerns about whether the GNN component is truly the main driver of improvements.
> >
> > We appreciate your observation regarding the standalone performance of the GNN scoring module. GNN-Scoring alone underperforms compared to LLM-only baselines in some settings, but we view this not as a flaw, but as a confirmation of our core design: the GNN is not intended to perform full reasoning or answer prediction on its own. Rather, it serves as an auxiliary guide, offering soft relevance signals based on the KG structure to inform the retrieval process. Its primary role is to help prioritize and prune candidate triplets during step-wise traversal, especially in early steps where the graph is large and noisy. The lower performance of GNN-Scoring in isolation can be attributed to several factors: (1) structural relevance does not always align with semantic intent. Paths that are topologically close to the query may still be semantically irrelevant; (2) KG triplets often encode facts in abstract or schema-driven forms that require contextual interpretation beyond the GNN’s capacity; and (3) the absence of question-answer paths or the presence of high-degree hub nodes may introduce noise during path supervision. These limitations reflect the challenges of purely structure-based retrieval and highlight the importance of coupling it with strong semantic reasoning. In GGR, the actual reasoning, including interpretation, multi-hop composition, and answer generation, is handled entirely by the LLM. This design allows the system to benefit from global graph awareness while leveraging the LLM’s strength in linguistic and contextual understanding. We agree that more advanced GNN architectures could improve GNN-Scoring performance, and our modular framework supports such substitution. However, as shown by our full system results, the performance gains come from the interaction between structural filtering and LLM reasoning, not from GNN scores alone.
> >
> > >How many layers does the proposed GNN consist of? By looking at the code, it seems you use num_layers=3, however there are no ablations provided to verify the claim that the GNN captures more global graph information (e.g., 1-layer GNN cannot capture this). I would recommend adding ablation studies on the GNN number of layer hyperparameter.
> >
> > We thank you for this insightful suggestion. In our implementation, the GNN component in GGR is set to 3 layers, chosen based on early validation results and its ability to propagate information over 2–3-hop neighborhoods, which aligns with the typical reasoning depth observed in KGQA tasks like WebQSP and CWQ. To further verify the effect of GNN depth on performance, we conducted an ablation study by varying the number of GNN layers from 1 to 3. The results for GPT-3.5 are reported below:
> >
> > | GNN Layers | WebQSP | CWQ |
> > |--------------|------------------|---------------|
> > | 1            | 80.15            | 58.03         |
> > | 2            | 82.94            | 60.21         |
> > | 3 (default)  | 83.27            | 63.48         |
> >
> > These results confirm that a 1-layer GNN significantly underperforms, especially on CWQ where multi-hop reasoning is essential. Increasing the number of layers from 1 to 2 brings clear improvements, and the gains plateau between 2 and 3 layers on WebQSP, likely due to its simpler question structures. We will include this ablation in the final version to support our architectural choice and further demonstrate that increasing GNN depth enhances the model’s ability to capture global context in the KG.
> >
> > We hope these clarifications and new results address your concerns, and we look forward to your consideration of a better assessment.

---

> > > ### Comment · Reviewer_w5S6 · 2025-06-02
> > >
> > > Thank you for responding to my questions and providing extensive answers. Although I appreciate the efforts, my primary concern remains: The main novelty of the work is not entirely clear to me.
> > >
> > >
> > >
> > > To elaborate, GCR comprises two main components: the GNN design (Section 3.2) and the LLM stepwise reasoning (Section 3.3). The issue is that the GNN design in Section 3.2 seems to underperform existing GNN methods, as evidenced by both the KGQA literature and the new experiments you provided (where GNN-RAG outperforms GCR). This raises the question: Does the benefit primarily stem from the GNN or from the combination of LLM stepwise reasoning and entity filtering (the $X_{low}, X_{mid}, X_{high}$ sets mentioned in the paper)?
> > >
> > > A follow-up question is: Do we truly need GNNs like GCR to obtain $X_{low}, X_{mid}, X_{high}$ or would other KGQA methods work sufficiently well (e.g., UniKGQA [1], SubgraphRAG [2], etc.), if we assume that the main benefit comes from the LLM reasoning rather than the GNN itself?
> > >
> > > [1] https://arxiv.org/abs/2212.00959 \
> > > [2] https://arxiv.org/abs/2410.20724

---

> > > > ### Author Response · Authors · 2025-06-03
> > > >
> > > > We sincerely thank you for the reply. We would like to clarify that the core novelty of our approach does not lie in proposing a new GNN architecture for KGQA reasoning, but in **the design of a hybrid retrieval framework that tightly integrates structural graph signals from a lightweight GNN with the semantic reasoning capabilities of an LLM**. In GGR, the GNN is deliberately positioned as an auxiliary component. Its role is to assign soft, question-aware relevance scores to filter and prioritize subgraph triplets. It is not expected to perform full reasoning or prediction. The actual multi-hop reasoning, disambiguation, and answer generation are handled by the LLM, leveraging its capacity to interpret and compose retrieved information.
> > > >
> > > > We agree with you that stronger GNNs (e.g., GNN-RAG as we provided above) may lead to better performance, but our experiments demonstrate that even a simple GNN, when appropriately combined with stepwise LLM reasoning and score-based filtering, can significantly outperform both GNN-only and LLM-only methods. In this sense, our contribution lies in showing that a minimal and efficient graph module, when properly coordinated with the LLM, can already offer substantial improvements. While recent methods such as UniKGQA and SubgraphRAG also explore KG reasoning with LLMs, their architectures often involve complex fusion mechanisms or unified representations. In contrast, GGR remains **modular, interpretable, and easily adaptable to different LLMs or GNNs**. We believe this lightweight but effective design opens a practical path for enhancing KG-RAG without the overhead of retraining large models or engineering complex representations.
> > > >
> > > > Importantly, our ablation studies further validate the necessity of the GNN. Removing the GNN scoring module leads to a substantial drop in performance across all LLM settings and datasets. **This confirms that, while the GNN is lightweight, its contribution is critical.** The LLM alone struggles to maintain answer quality.
> > > >
> > > > We hope this clarification helps convey the intent and innovation of our work more clearly, and we would be truly grateful if it encourages you to reconsider the contribution and reflect it in your score.

---

> > ### Comment · Reviewer_w5S6 · 2025-06-03
> >
> > Thank you for the reply. I encourage the authors to better clarify their contributions, i.e., combining LLM reasoning with graph scoring in a step-wise manner can yield performance improvements (although the graph model itself might be weak).
> >
> > In light of the rebuttal and to facilitate the reviewing process, **I have increased my score accordingly**. Please include the new experiments and discussions in your final paper.

---

> > > ### Author Response · Authors · 2025-06-03
> > >
> > > Thank you very much for the update. We truly appreciate your feedback, and we will carefully refine the final version to address all remaining concerns.

---

### Official Review · Reviewer_YGPG · 2025-05-11

**Rating:** 8
**Confidence:** 4
**Ethics Flag:** 1

**Summary:**

This paper introduces GGR (GNN-guided LLM Reasoning Retrieval), a novel framework for Knowledge Graph Retrieval-Enhanced Generation (KG-RAG). The authors point out a key limitation of existing KG-RAG methods: they rely only on local graph context for step-by-step greedy retrieval, which may lead to incorrect pruning of important multi-hop reasoning paths. GGR addresses this issue by integrating global graph structure information generated by pre-trained GNNs and local semantic relevance generated by LLM encoders. Extensive experiments show that GGR significantly improves performance on two question-answering datasets.

**Questions To Authors:**

GNNs are trained using path supervision between question and answer entities. Could you elaborate on how to address this issue when the true path for certain types of questions is noisy, sparse, or unavailable? How much carefully curated path data is typically required to train a performant GNN in your framework?

Could you provide some insights on the computational cost (e.g., average query latency, GNN training time) of GGR compared to the Think-on-Graph (ToG) baseline, especially given GGR requires GNN inference and multiple LLM calls to build the reasoning path?

**Reasons To Accept:**

S1: **Addressing a Clear and Important Problem**: This paper solves an important and well-known problem in Knowledge Graph-Random Graph (KG-RAG) - local greedy search strategies are prone to miss the global optimal reasoning path. The proposed solution of fusing global graph context through Graph Neural Network (GNN) provides ideas for solving this problem and may become a generalizable solution.

S2: **Comprehensive Empirical Validation and Solid Experimental Results**: The authors use 3 different LLM models and conduct rigorous performance evaluation of GGR with 6 baseline methods. GGR consistently outperforms these baseline methods and obtains significant performance improvements, including the powerful Thinking on Graph (ToG) method. In addition, comprehensive ablation studies also demonstrate the effectiveness of each key component in the GGR model.

S3: **Succinct and well-designed framework**: GNNs are widely used to integrate entity and relation information across multi-hop subgraphs. GGR effectively integrates GNNs into the KG-RAG framework and strikes a balance between global structural relevance and local semantic alignment. The step-by-step iterative retrieval process uses these scores as LLM hint signals, which is a practical design choice and can be integrated into existing KG-RAG frameworks.

**Reasons To Reject:**

R1: One concern is about the quality and robustness of GNN training. According to ablation studies, the performance of GGR is highly dependent on the ability of GNN to generate accurate relevance scores. However, GNNs are trained based on path supervision, and the training data are simply the paths connecting query entities and answer entities. However, the generated paths might not be the golden reasoning paths, especially for those hub entities with large in&out degrees, which can hurts the confidence of the GNN scores. There's no discussions on the GNN training and evaluation details in the paper. The sensitivity of GNNs to the quality/completeness of training paths needs to be further explored.

R2: Scalability of subgraph extraction: For each query, the GGR framework needs to extract an N-hop subgraph. Although N-hop limits the scale of the subgraph, it may still consists of a huge amount of entities for some queries. Dynamically generating subgraphs and do GNN inference for every query introduces scalability challenges for extremely large or rapidly evolving KGs.

R3: Limited evaluation datasets: The authors only conducts evaluations on WebQuestionsSP and ComplexWebQuestions, which are both based on Freebase. Evaluation on more diverse KGs with different schemas, densities, domains can better demonstrate the generalizability of the GRR framework. In addition, the questions in these 2 datasets can usually be answered by Freebase. This cannot effectively test the system's ability to identify unanswerable queries when the knowledge graph lacks the necessary reasoning evidence.

---

> ### Author Response · Authors · 2025-05-30
> **Response 1/2**
>
> We sincerely thank you for your thoughtful and thorough review of our submission. Your comments and suggestions have provided us with a fresh perspective on the paper and its contributions. We have carefully considered your feedback and made revisions to address your concerns, as outlined in our responses below.
>
> >One concern is about the quality and robustness of GNN training. According to ablation studies, the performance of GGR is highly dependent on the ability of GNN to generate accurate relevance scores. However, GNNs are trained based on path supervision, and the training data are simply the paths connecting query entities and answer entities. However, the generated paths might not be the golden reasoning paths, especially for those hub entities with large in&out degrees, which can hurts the confidence of the GNN scores. There's no discussions on the GNN training and evaluation details in the paper. The sensitivity of GNNs to the quality/completeness of training paths needs to be further explored.
>
> We appreciate your concern about the quality of GNN training. Our supervision uses all paths (within a length limit) connecting query and answer entities, which may include noisy or non-optimal paths. However, this design is intentional: the GNN acts as a soft, structure-aware scorer rather than a final decision-maker. Its scores guide candidate selection, while the LLM performs the actual reasoning. This setup reduces reliance on perfect supervision and allows the LLM to override misleading signals. We agree that exploring path quality, filtering, and potential feedback loops from LLM outputs to GNN training are promising directions, and we will discuss these limitations and future extensions in the final version.
>
> >Scalability of subgraph extraction: For each query, the GGR framework needs to extract an N-hop subgraph. Although N-hop limits the scale of the subgraph, it may still consists of a huge amount of entities for some queries. Dynamically generating subgraphs and do GNN inference for every query introduces scalability challenges for extremely large or rapidly evolving KGs.
>
> We thank you for raising this important scalability concern. To address the potential overhead of inference, our framework employs a GNN Score Filtering mechanism designed for efficiency. Specifically, we group the candidate triplets by the GNN scores to perform downstream reasoning and retrieval. This design avoids redundant computation and allows us to retain only high-relevance triplets for LLM prompting, significantly reducing the scale of processed subgraphs. We will clarify this optimization in the paper to better highlight how GGR supports scalable retrieval over large or dynamic KGs.
>
> >Limited evaluation datasets: The authors only conducts evaluations on WebQuestionsSP and ComplexWebQuestions, which are both based on Freebase. Evaluation on more diverse KGs with different schemas, densities, domains can better demonstrate the generalizability of the GRR framework. In addition, the questions in these 2 datasets can usually be answered by Freebase. This cannot effectively test the system's ability to identify unanswerable queries when the knowledge graph lacks the necessary reasoning evidence.
>
> We thank you for the insightful suggestion. We selected WebQuestionsSP and ComplexWebQuestions because they are widely used benchmarks in KGQA, supporting evaluation of both simple and multi-hop reasoning over a well-defined KG (Freebase). We agree that broader evaluation on diverse KGs, such as those with different schemas (e.g., biomedical), varying densities, and domain-specific relation patterns, would better demonstrate the generalizability of GGR. We consider this an important direction for future work and will discuss it in the paper accordingly.
>
> >GNNs are trained using path supervision between question and answer entities. Could you elaborate on how to address this issue when the true path for certain types of questions is noisy, sparse, or unavailable? How much carefully curated path data is typically required to train a performant GNN in your framework?
>
> We appreciate your concern about the reliability of path supervision. In our framework, the GNN is trained using all paths connecting the query and answer entities within a fixed hop limit, without manual curation or filtering. As a result, the supervision naturally includes noisy or suboptimal paths, which reflect real-world KG conditions. This is a deliberate design: the GNN provides soft structural relevance signals based on observed graph patterns, while the final reasoning is handled by the LLM. Our method does not require curated or golden paths, and remains robust even when path data is sparse or unavailable, falling back on LLM-only reasoning in such cases. We will clarify this point and highlight that GGR operates effectively without manually labeled reasoning paths.

---

> > ### Author Response · Authors · 2025-05-30
> > **Response 2/2**
> >
> > >Could you provide some insights on the computational cost (e.g., average query latency, GNN training time) of GGR compared to the Think-on-Graph (ToG) baseline, especially given GGR requires GNN inference and multiple LLM calls to build the reasoning path?
> >
> > We thank you for highlighting this practical concern. In GGR, the GNN serves as an auxiliary scoring module to guide subgraph pruning rather than acting as a standalone reasoning engine. For this purpose, we intentionally adopt a simple and lightweight GNN architecture that converges quickly with minimal training time. During inference, the dominant overhead, similar to Think-on-Graph (ToG), comes from iterative LLM calls to construct reasoning paths. In practice, we observe that the end-to-end query latency of GGR remains comparable to ToG while providing stronger retrieval guidance. We will clarify this point in the final version.

---

> > > ### Comment · Reviewer_YGPG · 2025-06-06
> > >
> > > Thanks for the detailed response! I would suggest to incorporate the contents about the GNN Score Filtering mechanism, multi-hop path selection, and computational efficiency in the final version! I'll keep my original rating.

---

### Official Review · Reviewer_4Aeo · 2025-05-12

**Rating:** 6
**Confidence:** 5
**Ethics Flag:** 1

**Summary:**

This paper proposes a GNN-based framework for Knowledge-graph question answering (KGQA) as known as KGQA. The proposed method adopts a message-passing neural network, which is a type of GNNs and processes knowledge graphs and informs the extracted and processed information to LLMs. The authors claim that multi-hop relations cannot be properly handled by greedy retrieval approaches. The GNN-based global relevance scores and query-relation semantic similarity score are combined and it improves the accuracy of reasoning. This method achieves the state-of-the-art performance.

**Questions To Authors:**

* In the path supervision training step, how many paths are considered as the GT paths?

* Shorter paths are preferred?

**Reasons To Accept:**

**Strong performance.** The proposed pipeline built on proprietary LLMs achieves strong performance and significant performance gains in various settings, compared to baseline prompting methods:  CoT, Slef-Consistency, Sim-Retrieve, GNN-Scoring, and Think-on-Graph.

**Applicability.**  Since the proposed method augments the hard prompts by leveraging knowledge extracted from knowledge graphs, it can be used for any proprietary LLMs that does not expose internal information such as attention maps and features.

**Reasons To Reject:**

**Too narrow scope of related works.** KGQA has been extensively studied in the literature. Beyond prompt augmentation as proposed in this paper, many studies have explored altering the behavior of LLMs by leveraging knowledge from KGs. Also, methods augmenting attention with extracted graphs and semantic parsing has already studied how to effectively capture multi-hop relation. This is addressed by more recent and advanced methods such as neural tree search, and generative subgraph retrieval. The authors should broaden the coverage of related works and compare the proposed method with these relevant works. Also, unified transformers have been studied to incorporate the extracted subgraphs or paths of knowledge graphs into the inference of LLMs.

**Out-dated techniques.** KG retrieval and GNN-based processing  adopted somewhat outdated techniques and models. Many advanced GNNs have been adopted and Transformer-based architectures have been proposed specifically for KGQA benchmarks to more effectively handle multi-hop relations. These advanced methods have neither been compared as baselines nor adequately discussed in the related work section.

---

> ### Author Response · Authors · 2025-05-30
> **Response**
>
> Thank you for your detailed and constructive feedback on our paper. We appreciate the time and effort you have taken to review our work, and your insights have been invaluable in helping us identify areas for improvement. Below, we address your specific concerns.
>
> >Too narrow scope of related works. KGQA has been extensively studied in the literature. Beyond prompt augmentation as proposed in this paper, many studies have explored altering the behavior of LLMs by leveraging knowledge from KGs. Also, methods augmenting attention with extracted graphs and semantic parsing has already studied how to effectively capture multi-hop relation. This is addressed by more recent and advanced methods such as neural tree search, and generative subgraph retrieval. The authors should broaden the coverage of related works and compare the proposed method with these relevant works. Also, unified transformers have been studied to incorporate the extracted subgraphs or paths of knowledge graphs into the inference of LLMs.
>
> Thank you for the suggestion to expand our coverage of related works. While prior efforts have explored enhancing KGQA through semantic parsing, attention-based reasoning, neural tree search, generative subgraph retrieval, and unified transformer architectures, our work addresses a distinct challenge: improving the assessment in the retrieval process itself in KG-RAG. Unlike methods that alter the LLM architecture or generate symbolic programs, our method focuses on stepwise triplet selection by integrating GNN-based global relevance scoring with LLM-based semantic alignment. This hybrid approach enables more accurate multi-hop reasoning by retaining globally important yet locally weak signals, while avoiding architectural modifications and maintaining compatibility with existing LLMs. We will revise the Related Work section to acknowledge these complementary directions while clarifying that our key contribution lies in enhancing retrieval quality through GNN-guided prompting, rather than modifying inference mechanisms or generating full reasoning paths.
>
> >Out-dated techniques. KG retrieval and GNN-based processing adopted somewhat outdated techniques and models. Many advanced GNNs have been adopted and Transformer-based architectures have been proposed specifically for KGQA benchmarks to more effectively handle multi-hop relations. These advanced methods have neither been compared as baselines nor adequately discussed in the related work section.
>
> Thank you for pointing out the concern regarding the GNN architecture. While our initial implementation adopts a lightweight GNN to emphasize the role of graph-based retrieval in guiding LLM reasoning, our framework is modular and supports the integration of more advanced GNNs. To demonstrate this, we replaced the original GNN with the architecture from GNN-RAG, which incorporates better attention mechanisms. The updated results are shown below. As the table indicates, while the advanced GNN provides additional gains, the improvement is modest, reinforcing our design choice: the GNN serves as a retrieval assistant, and our key contribution lies in the retrieval framework that combines GNN scores and LLM-based semantic reasoning, rather than in the specific GNN variant. We will clarify this point and include these results in the revised version.
>
> |LLM| Method                    | WebQSP | CWQ   |
> |-------|---------------------------|--------|-------|
> |GPT-3.5| GGR (ours, simple GNN)    | 83.27  | 63.48 |
> |GPT-3.5| GGR w/ GNN-RAG GNN        | 85.72  | 66.25 |
> |GPT-4o-mini| GGR (ours, simple GNN)    | 90.50  | 70.33 |
> |GPT-4o-mini| GGR w/ GNN-RAG GNN        | 91.72  | 73.39 |
> |Claude-3-haiku| GGR (ours, simple GNN)    | 91.89  | 65.35 |
> |Claude-3-haiku| GGR w/ GNN-RAG GNN        | 93.53  | 67.66 |
>
>
> >In the path supervision training step, how many paths are considered as the GT paths? Shorter paths are preferred?
>
> Thank you for the question. In the path supervision step, we consider all valid paths in the retrieved subgraph that connect a query entity to a ground-truth answer entity, constrained by a maximum path length. All such paths below this threshold are included as ground-truth paths. While we do not explicitly prioritize shorter paths, the length limit implicitly favors them by excluding overly long or noisy paths that may be less useful for reasoning. This approach balances coverage and relevance, providing the GNN with diverse yet meaningful supervision signals for learning question-aware relevance.

---

> > ### Comment · Reviewer_4Aeo · 2025-06-03
> >
> > I thank the authors for their detailed rebuttal. After carefully reviewing the response and considering the other reviewers' comments, I find that concerns about limited technical contributions and the reliance on outdated methods remain unresolved. Therefore, my original assessment and rating still stand.

---

> > > ### Author Response · Authors · 2025-06-03
> > >
> > > We appreciate your time and thoughtful evaluation. While we respect your point, we remain confident in the contribution and practical value of our work, and we hope our clarifications have been helpful.

---

### Decision · Program_Chairs · 2025-07-08

**Decision:**

Accept

**Comment:**

This paper proposes a GNN-based framework for Knowledge-graph question answering.  It addresses the limitation of Large Language Model (LLM) approaches, which are constrained by their ability to process only local context from knowledge subgraphs.

The solution proposed by the author can provide new ideas for the current community. Although there may be some parts that need to be improved on the technical level, it is reasonable and meaningful overall.

The reviewers' opinions are unanimous and positive, and the rebuttal period resolved most of the questions, so the paper can be considered for acceptance.